# Clinical effectiveness and components of Home-pulmonary rehabilitation for people with chronic respiratory diseases: a systematic review protocol

Md Nazim Uzzaman ![ORCID],[1] Soo Chin Chan,[2] Ranita Hisham Shunmugam,[3] Julia Patrick Engkasan,[2] Dhiraj Agarwal,[4] G M Monsur Habib,[1,5] Nik Sherina Hanafi,[2] Tracy Jackson,[1] Paul Jebaraj,[6] Ee Ming Khoo,[2] Su May Liew,[2] Fatim Tahirah Mirza,[7] Hilary Pinnock,[1] Roberto A Rabinovich[8,9]

For numbered affiliations see end of article.

**Correspondence to**
Dr Roberto A Rabinovich;
roberto.rabinovich@ed.ac.uk

## ABSTRACT

**Introduction** Chronic respiratory diseases (CRDs) are common and disabling conditions that can result in social isolation and economic hardship for patients and their families. Pulmonary rehabilitation (PR) improves functional exercise capacity and health-related quality of life (HRQoL) but practical barriers to attending centre-based sessions or the need for infection control limits accessibility. Home-PR offers a potential solution that may improve access. We aim to systematically review the clinical effectiveness, completion rates and components of Home-PR for people with CRDs compared with Centre-PR or Usual care.

**Methods and analysis** We will search PubMed, CINAHL, Cochrane, EMBASE, PeDRO and PsycInfo from January 1990 to date using a PICOS search strategy (Population: adults with CRDs; Intervention: Home-PR; Comparator: Centre-PR/Usual care; Outcomes: functional exercise capacity and HRQoL; Setting: any setting). The strategy is to search for 'Chronic Respiratory Disease' AND 'Pulmonary Rehabilitation' AND 'Home-PR', and identify relevant randomised controlled trials and controlled clinical trials. Six reviewers working in pairs will independently screen articles for eligibility and extract data from those fulfilling the inclusion criteria. We will use the Cochrane risk-of-bias tool and Grading of Recommendations Assessment, Development and Evaluation (GRADE) approach to rate the quality of evidence. We will perform meta-analysis or narrative synthesis as appropriate to answer our three research questions: (1) what is the effectiveness of Home-PR compared with Centre-PR or Usual care? (2) what components are used in effective Home-PR studies? and (3) what is the completion rate of Home-PR compared with Centre-PR?

**Ethics and dissemination** Research ethics approval is not required since the study will review only published data. The findings will be disseminated through publication in a peer-reviewed journal and presentation in conferences.

**PROSPERO registration number** CRD42020220137.

## Strengths and limitations of this study

► A systematic review of the effectiveness, completion rates and components of Home-pulmonary rehabilitation (Home-PR) for chronic respiratory diseases (CRDs) is needed to inform patients and providers especially when healthcare accessibility is restricted by geography, demography or during pandemics.

► The review methods are in accordance with Cochrane methodology and Preferred Reporting Items for Systematic Reviews and Meta-Analyses publishing guidelines.

► Issues like heterogeneity, poor reporting of published trials may affect confidence in results although we expect to provide robust evidence supporting the successful implementation of Home-PR services for people with CRDs.

► The multi-disciplinary, multinational research team will enable a nuanced interpretation of the findings.

## INTRODUCTION

Chronic respiratory diseases (CRDs) including chronic obstructive pulmonary disease (COPD), remodelled asthma, pulmonary impairment after tuberculosis (PIAT), interstitial lung disease (ILD), bronchiectasis and cystic fibrosis (CF), among others, affect an estimated 545 million people globally.[1] Around 4 million people die prematurely from CRDs each year across the world,[2] and COPD, asthma, and tuberculosis are among the top 30 conditions that cause high rates of disability-adjusted life-years.[3] CRDs, in particular COPD, are associated with breathlessness, fatigue and muscle dysfunction which contribute to reduced physical activity levels and functional exercise capacity.[4] Independent of the severity of airway obstruction, this functional impairment is related to decreased health-related quality-of-life (HRQoL), increased adverse events and mortality.[5 6]

Pulmonary rehabilitation (PR) is an essential component of CRD care[7] that improves functional exercise capacity, HRQoL and reduces the burden of chronic respiratory symptoms.[8 9] It is defined as a comprehensive,

multidisciplinary and multifaceted intervention based on a thorough patient assessment, followed by individually tailored therapies that are designed to improve the physical and psychological conditions of people with CRDs and to support the long-term adherence to health-enhancing behaviours.[10 11] The components of PR include exercise training, education, nutritional support, smoking cessation, lifestyle modification and self-management, among others. PR is indicated for patients who continue to experience symptoms despite optimising pharmacological treatment.[12 13]

Despite proven effectiveness,[11 14 15] PR is under-used. The reasons for poor attendance and completion rates are multifactorial and commonly identified barriers include: low referral rate; inconvenient timing of the programmes necessitating time off work; geographical distance to PR centres which can be made worse in some countries by poor transport infrastructure.[16–20] While pertinent even in high-income countries[21–23] many of these barriers are exacerbated in low/middle-income countries (LMICs) where there is a lack of structured PR facilities especially in rural communities.[24 25]

Typically, PR is provided in hospital centres (Centre-PR),[26] but globally different models are tailored to the local context such as Community-PR,[27] and Home-PR with telephone-mentoring,[28] or telerehabilitation programmes.[29] The ongoing COVID-19 pandemic has added strain to PR services by increasing the population for whom PR is indicated and adding barriers to the delivery of the treatment due to cross-infection issues. There is, therefore, an increased interest in Home-PR[30] as a strategy to overcome these barriers. A Cochrane review of 65 studies (3822 patients) has established the effectiveness of standard Centre-PR programmes in COPD[14] with a subgroup analysis suggesting that PR delivered in a hospital centre may have a greater treatment effect than PR delivered in the community/home. Using the same definitions, three reviews (Wuytack *et al*,[31] Chen and Xiao-XiaoYang[32] and Neves *et al*[33]) included studies comparing PR delivered in different settings and both concluded that Home/Community-PR could be as effective as Centre-PR for people with COPD. Combining home and community services, however, overlooks the distinction between a community-based group supervised in person by a healthcare professional and a programme delivered to an individual in their own home. These reviews are also limited by disease (COPD only), although there is evidence that PR is of benefit in bronchiectasis[11] and ILD.[34] Taito *et al* in a scoping review also included people recovering from COVID-19.[35]

A recently published Cochrane review assessed the effectiveness and safety of telerehabilitation for people with CRDs when compared with Centre-PR or no rehabilitation[36] and concluded that primary or maintenance PR telerehabilitation achieved similar outcomes to Centre-PR. In this review, remote delivery of PR was defined by the use of telecommunications technology to deliver PR services to individuals or groups (either physical or virtual) in any location, including in the patient's home or at a healthcare centre. In contrast, in our review, the definition of Home-PR is that the sessions are undertaken by an individual by themselves (though a family member may be involved) and typically at home. Apart from baseline and post-PR assessments,[35] the patient does not attend a centre (either a hospital centre or a local 'satellite' centre) and is not supervised face-to-face by a healthcare professional (though there may be remote communication from a healthcare professional for some or all of the session).

An additional distinction is that we defined Home-PR as comprising both exercise and at least one non-exercise component for a duration of not less than 4 weeks. This contrasts with other reviews[32 36 37] that included exercise training programmes (ie, without the non-exercise component that is normally included in Centre-PR[8]). These reviews did not seek to identify components with greater impact on positive patient outcomes. We therefore aim to systematically review the literature to assess the effectiveness, completion rates and components used in effective Home-PR for people with CRDs.

## OBJECTIVES
In people with CRDs, we will:
1. Assess the clinical effectiveness of Home-PR (see table 1 for definition) compared with Centre-PR or Usual care at improving health outcomes (ie, exercise capacity (primary outcome), HRQoL (primary outcome), dyspnoea, muscle fatigue, exacerbations and hospitalisations for CRD).
2. Describe the components of Home-PR that are associated with successful interventions (eg, intensity of exercise, duration of the programme, education and/or other non-exercise components, frequency of supervision, information/resources, involvement of family members).
3. Compare the completion rate (defined as participating in at least 70% of PR sessions) of Home-PR with Centre-PR.

## METHODS AND ANALYSIS
We will follow Cochrane methodology,[37] and use Preferred Reporting Items for Systematic Reviews and Meta-Analyses (PRISMA) guidelines[38] to report our review findings. The review is registered with PROSPERO, any changes to the published record will be reported.

### Search strategy
We will develop a search strategy, including disease-specific search terms, and identify records through searching the following databases: PubMed, CINAHL, Cochrane, EMBASE, PeDRO and PsycInfo (online supplemental appendix 1). The strategy will search for 'Chronic Respiratory Disease' AND 'Pulmonary Rehabilitation' AND 'Home-PR' from 1990, when global COPD

**Table 1** PICOS table for the search strategy

| PICOS | Description, inclusion | Exclusion criteria | Operational rules |
|---|---|---|---|
| Population | ▲ Adults with primary diagnosis of chronic respiratory diseases (CRDs).<br>▲ Age >18 years.<br>▲ Comorbidity will not be an exclusion criterion. | ▲ Pregnant women and paediatric population.<br>▲ Rehabilitation provided to predominant condition is non-respiratory conditions.<br>▲ Recovery from acute infections or injury (eg, immediately post-COVID) until the condition has been stable for 6 months.<br>▲ Conference abstract.<br>▲ Lung cancer.<br>▲ Pulmonary hypertension. | PR delivered to people with CRDs such as chronic obstructive pulmonary disease (COPD), remodelled asthma, pulmonary impairment after tuberculosis (PIAT), bronchiectasis, interstitial lung disease (ILD), cystic fibrosis (CF), stable post-COVID (but excluding post-intensive care unit rehabilitation) will be studied. We will also include PR delivered to people with more than one CRD, or undifferentiated chronic respiratory conditions. |
| Intervention | Home-pulmonary rehabilitation (PR) which comprises both exercise and at least one non-exercise component for a duration not lesser than 4 weeks. | Formal hospital or community medical centre-based programmes. | 'Home-PR'—the key criterion is that the sessions are undertaken by an individual by themselves (though a family member may be involved) and typically at home. Apart from baseline and post-PR assessments,[35] the patient does not attend a centre (either a hospital centre or a local 'satellite' centre) and is not supervised face-to-face by a healthcare professional (though there may be remote communication from a healthcare professional for some or all of the session). Exercise sessions typically include aerobic, endurance, resistance and reconditioning exercises, though local resources and preferences may include other exercise modalities. Non-exercise components commonly include patient education, energy conservation training, smoking cessation, psychological support, self-management skill development or other recognised PR interventions along with optimisation of pharmacotherapy. |
| Comparison | Either population receiving 'Centre-PR' or receiving 'Usual care'. | No control groups. | 'Centre-PR'—the key criterion is that the sessions are under direct healthcare professional's supervision. The 'Centre' can be in a hospital, community setting, or remote facility. Centre-based sessions are normally group-based (though it is recognised that this may be modified in the context of a pandemic). Telehealth services where patients attend a supervised satellite Centre would be considered as Centre-PR.<br>'Usual care'—is the standard care received by individual with CRD in the relevant healthcare system but excluding the exercise components of PR. |
| Outcomes | Consist of either one of the following outcome measures<br>▲ Health-related quality of life (HRQoL).<br>▲ Functional exercise capacity.<br>±<br>Additional outcome(s)<br>▲ Uptake of the service, completion rates.<br>▲ Assessment of motivation/others intermediate outcome.<br>▲ Activities of daily living.<br>▲ Physical activity level.<br>▲ Symptom control.<br>▲ Psychological status.<br>▲ Healthcare burden, for example, exacerbation rates, hospitalisation, etc.<br>▲ Adverse effect. | Not including HRQoL or any measurement of exercise capacity as outcome. | Validated instruments will be considered:<br>▲ HRQoL: St Georges Respiratory Questionnaire (SGRQ), Chronic Respiratory Questionnaire (CRQ), EuroQol Five Dimension (EQ-5D).<br>▲ Functional exercise capacity: 6-minute walk test (6MWT), incremental shuttle walking test (ISWT), endurance shuttle walking test (ESWT). We will also include step tests and sit-to-stand tests that are sometimes used in Home-PR assessments.[45]<br>▲ Symptom control: Modified Medical Research Council (mMRC), Clinical COPD Questionnaire (CCQ), Borg scale.<br>▲ Psychological status: Hospital Anxiety and Depression Scale (HADS), Patient Health Questionnaire-9 (PHQ-9), State-Trait Anxiety Inventory (STAI), Beck Inventory test. |
| Setting | Any settings | | Low or high resource settings irrespective of level of economies of the countries. |
| Study designs | Randomised controlled trials (RCTs); clinical controlled trials (CCTs). | Cohort study, case series, case report. | We will exclude studies that do not have a control group. We will consider RCTs to answer all of the three research questions (ie, (1) effectiveness, (2) components and (3) completion rate of Home-PR), and consider CCTs to answer research questions 2 and 3. |
| Language | No language restriction. | | |

guidelines first recommended PR.[39] We will check reference lists and conduct forward citation on included studies and on Cochrane reviews of PR.[14 17] We will not impose any language restriction, and will arrange for translation to English to enable selection and data extraction.

## Selection process

We will select studies that compare Home-PR for people with CRDs with Centre-PR and/or Usual care (see definitions and details of our PICOS (Population, Intervention, Comparison, Outcomes, Setting) criteria in table 1). Following training on 100 randomly selected records, six reviewers working in pairs (MNU and TJ, JPE and FTM, DA and PJ) will duplicate screen titles and abstracts and identify potentially eligible studies. Disagreements will be resolved by discussion with the review team (HP, RAR, SML, GMMH, NSH and SCC) as necessary. After retrieval of the full text of potentially eligible studies, the six reviewers working in the same pairs will independently screen the studies against the selection criteria. Disagreements will be resolved by discussion within the team to arbitrate and determine rules for operationalising the inclusion/exclusion criteria. Anything that remains unclear, will be clarified by contacting the authors; if this fails, the study will be listed as 'potentially relevant study'. All processes will be reported in a PRISMA flow diagram,[38] and excluded full-text papers will be tabulated with reasons for exclusion.

## Outcome measurement

Our primary outcomes will be HRQoL and functional exercise capacity. We are interested in preassessment and postassessment or if an immediate post is not provided, the nearest figure to that. See table 1 for details and description of secondary outcomes.

## Data management and extraction

We will develop a customised data extraction form based on Cochrane Effective Practice and Organisation of Care (EPOC) guidance.[40] This form will be piloted by all the researchers to standardised use and revised to endure that it captures all relevant information including the PICOS criteria, definitions used and outcome measurements. Data extraction will be carried out by six reviewers working in pairs (MNU, TJ, JPE, FTM, DA, PJ). General information such as date of extraction, name of the reviewer, article title, trial eligibility including type of study, participants, methods, number of participants in each group, reference of trials, intervention group, cointerventions, serious adverse events, description of funding, ethical approval will be extracted from included full-text papers. We will contact authors for any missing data. If this is not possible and the missing data seem to introduce serious bias, we will perform sensitivity analysis of the impact of including such studies in the overall assessment of results.

## Risk of bias assessment

Methodological quality of all included randomised controlled trials (RCTs) will be assessed independently by reviewers (MNU, TJ, JPE, FTM, DA, PJ working in pairs) using the 'Cochrane Risk of Bias' tool.[37] Discrepancies will be resolved by discussion with the team. We will assess the papers for selection, performance, detection, attrition, reporting and other sources of bias, and assess the overall risk of bias. We will record and tabulate a summary of the assessment with the overall judgement. To assess the risk of bias of clinical controlled trials, we will use the 'Risk of Bias in Non-randomised Studies of Interventions' tool.[41] We will include all studies in our primary analysis but take into account the risk of bias of the studies when considering the intervention effects. If there are sufficient studies we may undertake sensitivity analyses omitting studies at high risk of bias.

## Heterogeneity and reporting bias

We will assess and investigate reasons for any heterogeneity using the $I^2$ statistic[42] and create a funnel plot to test for publication bias[43] unless we have fewer than 10 trials.

## Data analysis
### Objective 1

We plan to undertake meta-analysis for the primary outcomes and some secondary outcomes (eg, HRQoL, dyspnoea, muscle fatigue, exacerbations, hospitalisations), comparing Home-PR first with Usual care and then with Centre-PR. Heterogeneous outcomes for which a meta-analysis is inappropriate will be synthesised narratively. For homogenous data from RCTs, we will perform a pooled quantitative synthesis using an inverse variance method and a random-effects model in the meta-analysis. We will consider pooled mean differences if the same outcome measurement tool is used in the included RCTs. However, if (as expected) outcome measurement tool varies among trials, we will consider standardised mean differences (SMDs). Our hypothesis is that Home-PR is non-inferior to Centre-PR, but a clinically meaningful non-inferiority margin cannot be inferred using SMDs. If sufficient studies use the same measure for functional exercise capacity or health-related quality of life, we will define the non-inferiority margin as the minimum clinically important difference. We will use Review Manager software (RevMan 2020, V.5.4.1) to perform meta-analysis.

### Objective 2

The components of Home-PR will be described and a matrix compiled to identify any associations with successful interventions.

### Objective 3

We will use a narrative approach to synthesise completion in Home-PR and Centre-PR groups. If sufficient studies report completion rates, we will consider a sub-group analysis based on the threshold of 70% completion.

## Subgroup analyses

Depending on the papers included, we will perform subgroup analyses. Subgroups may include high/LMICs, CRD diagnosis (eg, COPD, ILD, bronchiectasis, stable post-COVID lung disease, mixed CRD), severity as defined in internationally recognised guidelines,[12] intensity of intervention (number of weeks, sessions per week, workload, completion rate) and arrangements for supervision of the PR programme.

## Interpretation of findings

We will use the Grading of Recommendations Assessment, Development and Evaluation (GRADE) approach[44] to assess the quality of evidence and strength of recommendations for the primary outcomes and the important secondary outcomes (listed in table 1).

## Patient and public involvement

Patients who are involved in the RESPIRE programme of work on PR have endorsed the importance of Home-PR for improving accessibility to rehabilitation. They will be involved in interpreting the findings and the implications for intervention development and the overall programme of work

## DISCUSSION

Home-PR has particular resonance at the time of developing this protocol because of the COVID-19 pandemic which has resulted in Centre-PR services being halted. More generally, there is an interest in offering Home-PR as a strategy to overcome the practical barriers of time and distance and increase the accessibility of PR services especially in LMICs. There are, however, concerns that the relative lack of supervision and the loss of peer group support may reduce effectiveness. Hence a review on the effectiveness of Home-PR and its components is timely to inform patients, professionals and healthcare service providers considering Home-PR options.

## ETHICS AND DISSEMINATION

This systematic review protocol will use publicly available data without direct involvement of human participants. Therefore, approval from an ethics committee is not essential. We will present our review findings at national and international scientific meetings and conferences, and publish in a peer-reviewed journal. In addition, we will use innovative dissemination strategies including virtual seminars and social media.

## Author affiliations
[1]NIHR Global Health Research Unit on Respiratory Health (RESPIRE), Usher Institute, The University of Edinburgh, Edinburgh, UK
[2]Faculty of Medicine, University of Malaya, Kuala Lumpur, Malaysia
[3]Library, University of Malaya, Kuala Lumpur, Malaysia
[4]KEM Hospital and Research Centre, Pune, India
[5]Community Respiratory Centre, Bangladesh Primary Care Respiratory Society, Khulna, Bangladesh
[6]Christian Medical College, Vellore, India
[7]Faculty of Health Sciences, Universiti Teknologi MARA, Selangor, Malaysia
[8]Respiratory Department, Royal Infirmary of Edinburgh, Edinburgh, UK
[9]Centre for Inflammation Research, QMRI, The University of Edinburgh, Edinburgh, UK

**Acknowledgements** We are grateful to the RESPIRE patient colleagues who have offered their advice on the pulmonary rehabilitation programme work.

**Contributors** RAR and HP led the team (MNU, SCC, RHS, JPE, DA, GMMH, NSH, TJ, PJ, EMK, SML, FTM) who all contributed to the development of the protocol. MNU drafted the first version of the manuscript with support from SCC and DA, which was revised with contributions from all the authors. All authors have critically reviewed and approved the final manuscript.

**Funding** This research was commissioned by the UK National Institute for Health Research (NIHR) Global Health Research Unit on Respiratory Health (RESPIRE): 16/136/109, using UK Aid from the UK Government. The RESPIRE collaboration comprises the UK Grant holders, Partners and research teams as listed on the RESPIRE website (www.ed.ac.uk/usher/respire), including Harish Nair. TJ is part-funded by the NIHR RESPIRE. GMMH has an NIHR RESPIRE PhD studentship. MNU, DA, SML hold NIHR RESPIRE Fellowships. HP, RAR, EMK, SCC, JPE, NSH, SML are co-investigators of NIHR RESPIRE-funded PR feasibility studies in their respective Centres. RESPIRE is funded by the National Institute of Health Research using Official Development Assistance (ODA) funding.

**Disclaimer** The views expressed in this publication are those of the author(s) and not necessarily those of the NIHR or the UK Department of Health and Social Care.

**Competing interests** GMMH owns a Pulmonary Rehabilitation clinic in Bangladesh. All other authors declare no competing interests.

**Patient consent for publication** Not applicable.

**Provenance and peer review** Not commissioned; externally peer reviewed.

**ORCID iD**
Md Nazim Uzzaman http://orcid.org/0000-0001-9528-6299

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
