## [Reviewer comments · BMJ Open]

ARTICLE DETAILS

TITLE (PROVISIONAL)	Clinical effectiveness, and components of home-pulmonary rehabilitation for people with chronic respiratory diseases: A systematic review protocol
AUTHORS	Uzzaman, Md. Nazim; Chan, Soo Chin; Ranita, Hisham; Engkasan, Julia; Agarwal, Dhiraj; Habib, GM Monsur; Hanafi, Nik Sherina; Jackson, Tracy; Jebaraj, Paul; Khoo, Ee Ming; Liew, Su May; Mirza, Fatim; Pinnock, Hilary; Rabinovich, Roberto

VERSION 1 – REVIEW

REVIEWER	Lewis, Adam Imperial College London, Department of Respiratory Epidemiology, National Heart and Lung Institute
REVIEW RETURNED	10-May-2021

GENERAL COMMENTS	Thank you to the authors for this submission. I read it with great interest. The need for home-based PR has become increasingly important in the context of COVID-19, and also the importance of structured availability of PR in lower-middle income countries is well stated. Your manuscript is brief and you have plenty more words to play with and so I have made some suggestions below which will hopefully help in you considering some amendments to your work. The abstract is clear and details databases, PICO focus, quality assessment and research questions. It may be beneficial to add example search terms and types of studies included (?RCT only) within the abstract. Introduction You need a paragraph in your introduction of similar research performed in the field already. In the introduction it may be worth referencing the recent Cochrane review in telerehabilitation(1) and maybe the work by Taito et al(2), and Wuytack et al(3) illustrating the differences in your methods and highlighting/increasing the evidence for the gaps for you to address. There may be others too. A definition of what home-based PR is would be good within the main text of the review, considering it appears that some telehealth programmes will and will not be included in your table. Also it sometimes can only be called home PR if a centre based assessment is performed. This needs to be made clear. List advantages and disadvantages of including all Chronic Respiratory Diseases as the population vs respiratory disease specific populations separately. From a clinical perspective it is
---

likely all CRD patients included would be participating in a programme together so that is relevant and applicable. However, methodologically it may cause some issues when comparing results. Will you perform a sub-group or sensitivity analysis per respiratory condition?

Objectives

I really like the ambition of the multi-faceted objectives here, all of which are important. Good to define completion rate too. Does this mean that trials using a lower completion rate than this will be excluded from the analysis, like Holland et al(4). Alternatively you could lower your threshold for completion as has been the case in previous studies, and relates to the improvements seen after 4 weeks(5).

Ref 35: An updated 2020 PRISMA statement is available and recommended to follow on the PRISMA website and in this paper: The PRISMA 2020 statement: an updated guideline for reporting systematic reviews | The BMJ

Line 50: In your search strategy please report that you will include disease specific search terms, even if you don't list them all, as you do in the Pubmed term table

Line 52: state hospital centres – community based approached are often in centres too (like leisure centres) and this may help differentiate.

Search strategy:

Depending on your research questions it may be appropriate to include different study designs, so RCTs would be appropriate to study clinical effectiveness but other study designs wouldn't. Determining the completion rates and frequency or matrix of components may enable you to look at other research designs. For example Nolan et al(6) performed a propensity matched cohort study comparing completion rates which may be useful to consider but would currently be excluded. This may also be useful to consider: <https://www.lungindia.com/article.asp?issn=0970-2113;year=2021;volume=38;issue=3;spage=211;epage=215;aulast=Priya>

You may also wish to follow up the authors of this paper to see if publication is planned:

https://www.atsjournals.org/doi/pdf/10.1164/ajrccm-conference.2021.203.1_MeetingAbstracts.A4148

Therefore, I think it is worth stating for each research question which study designs would be considered. I appreciate this information is in Table 1. But I think warrants reiterating specifically per research questions.

If you are interested in PR in stable-state COVID-19 patients you should include such terms in your search terms like COVID19 OR COVID-19 OR SARS-CoV-2. In the methods a description of what a chronic respiratory disease is may help justify COVID-19 as a chronic disease, because this could also be debated. See this article which shows persistent changes at 12 months post COVID [https://www.thelancet.com/journals/lanres/article/PIIS2213-2600\(21\)00174-0/fulltext](https://www.thelancet.com/journals/lanres/article/PIIS2213-2600(21)00174-0/fulltext)

	However, individuals with COVID-19 may require a different PR due to different educational needs and exercise prescription potentially, also needing outcome measures to be validated, so I'm not sure results from such trials could be compared with other respiratory diseases. Perhaps a sub-group analysis? Why are six reviewers required? Often only two required with a third to discuss discrepancies. Also why in pairs and what does the training of the 100 randomly selected articles involve, How will this be random? This is interesting as I haven't read about this method of reviewing with such a large team and understand it will have its merits but may also have some limitations. These would be important to discuss so others could consider whether valuable for their reviews. If this method has been used elsewhere, please reference. If not it may be novel, which isn't necessarily a bad thing. Outcome measurement: Provide some more justification for having more than 1 primary outcome measure. Most home-PR Trials previously have included a centre-based assessment and so will have a standardised evidenced based outcome measure (6MWT, ISWT, ESWT)(4, 6-9) so this may be a justification you can provide for choosing this as a primary outcome. Without such justification some clinicians may feel that such tests are often not used clinically in home based PR programmes because they are difficult to perform and others may be preferred such as the Sit to Stand Tests. These however have not been widely validated for exercise prescription purposes(10). Although a centre-based assessment is deemed an essential component of a PR programme(11) You could add step tests and sit to stand tests (5repSTS, 1 minSTS) into your table of exercise capacity tests to allow for such home-based transferable outcomes to be analysed, should your definition of home-based or telehealth programmes including those without a centre-based assessment. Health related quality of life questionnaires can definitely be completed in the home or centre based environments. All three of your outcomes are really important and clinically relevant Outcome one: what is the statistical test or final outcome expected ? pooled mean differences, standardised mean differences. Also state what models you are considering for meta-analysis. Consider how you will account for missing data in trials from reporting quality (possibly in your secondary outcome measures this may occur). What software will you use to perform the meta-analysis? Will usual care, and centre-based control group study outcomes be pooled? Outcome 2: This paper may be a good read regarding how to explore the value of various components within a PR intervention(12)
--	---

	Outcome 3: How will this be analysed? Will an overall mean completion rate be presented or will you compare this means with a statistical test? chi-squared test? Risk of Bias: Will all studies be included in quantitative analyses regardless of quality? How will you assess the risk of bias in non randomised controlled trials? Consider ROBINS-I(13) PPI “Patients are involved in the with the RESPIRE programme of work on pulmonary rehabilitation.” Needs rewording Other The statement in your definition for home-PR requires a little more justification: “The patient does not attend a Centre, is not directly supervised by a healthcare professional (though they may be monitored remotely for some or all of the session).” In my experience telehealth PR is often lead by a clinician remotely, which implies supervision rather than monitoring. Is there a difference with monitoring that means no direct communication is made between the clinician and patients perhaps? In the definition of Centre-based PR you state that telehealth programmes are included in this arm too, but if patients attend a centre. Why would they need to attend a centre in order to do telehealth? Is the main important difference here that patients are exercising in a group? This addition may help add clarity too. Hope this helps and good luck in your revision. References  1. Cox NS, Dal Corso S, Hansen H, McDonald CF, Hill CJ, Zanaboni P, et al. Telerehabilitation for chronic respiratory disease. Cochrane Database Syst Rev. 2021;1:Cd013040. 2. Taito S, Yamauchi K, Kataoka Y. Telerehabilitation in Subjects With Respiratory Disease: A Scoping Review. Respir Care. 2021;66(4):686-98. 3. Wuytack F, Devane D, Stovold E, McDonnell M, Casey M, McDonnell TJ, et al. Comparison of outpatient and home-based exercise training programmes for COPD: A systematic review and meta-analysis. Respirology. 2018;23(3):272-83. 4. Holland AE, Mahal A, Hill CJ, Lee AL, Burge AT, Cox NS, et al. Home-based rehabilitation for COPD using minimal resources: a randomised, controlled equivalence trial. Thorax. 2017;72(1):57-65. 5. Sewell L, Singh SJ, Williams JE, Collier R, Morgan MD. How long should outpatient pulmonary rehabilitation be? A randomised controlled trial of 4 weeks versus 7 weeks. Thorax. 2006;61(9):767-71. 6. Nolan CM, Kalaraju D, Jones SE, Patel S, Barker R, Walsh JA, et al. Home versus outpatient pulmonary rehabilitation in COPD: a propensity-matched cohort study. Thorax. 2019;74(10):996-8. 7. Horton EJ, Mitchell KE, Johnson-Warrington V, Apps LD, Sewell L, Morgan M, et al. Comparison of a structured home-based rehabilitation programme with conventional supervised pulmonary rehabilitation: a randomised non-inferiority trial. Thorax. 2018;73(1):29-36.
--	--

	8. Hansen H, Bieler T, Beyer N, Kallemsen T, Wilcke JT, Østergaard LM, et al. Supervised pulmonary tele-rehabilitation versus pulmonary rehabilitation in severe COPD: a randomised multicentre trial. Thorax. 2020;75(5):413-21. 9. Tsai LL, McNamara RJ, Moddel C, Alison JA, McKenzie DK, McKeough ZJ. Home-based telerehabilitation via real-time videoconferencing improves endurance exercise capacity in patients with COPD: The randomized controlled TeleR Study. Respirology. 2017;22(4):699-707. 10. Houchen-Wolloff L, Daynes E, Watt A, Chaplin E, Gardiner N, Singh S. Which functional outcome measures can we use as a surrogate for exercise capacity during remote cardiopulmonary rehabilitation assessments? A rapid narrative review. ERJ Open Res. 2020;6(4). 11. Holland AE, Cox NS, Houchen-Wolloff L, Rochester CL, Garvey C, ZuWallack R, et al. Defining Modern Pulmonary Rehabilitation. An Official American Thoracic Society Workshop Report. Ann Am Thorac Soc. 2021;18(5):e12-e29. 12. Machado A, Matos Silva P, Afreixo V, Caneiras C, Burtin C, Marques A. Design of pulmonary rehabilitation programmes during acute exacerbations of COPD: a systematic review and network meta-analysis. Eur Respir Rev. 2020;29(158). 13. Sterne JA, Hernán MA, Reeves BC, Savović J, Berkman ND, Viswanathan M, et al. ROBINS-I: a tool for assessing risk of bias in non-randomised studies of interventions. BMJ. 2016;355:i4919.
--	--

REVIEWER	Layton, Aimee Columbia University Medical Center, Department of Pediatrics
REVIEW RETURNED	17-May-2021

GENERAL COMMENTS	This manuscript describes the protocol and rationale for performing a thorough review of the current literature on the effectiveness of home pulmonary rehabilitation compared to center based. The rationale, methods and discussion are clear and well supported. I have no major comments.
---

REVIEWER	BOREL, Jean INSERM U1042, HP2 Laboratory
REVIEW RETURNED	28-May-2021

GENERAL COMMENTS	Thank you for inviting me to review this manuscript. Dr Uzzaman is currently conducting a systematic review with meta-analysis to evaluate the effectiveness, completion rates and components of home-based versus centre-based rehabilitation programmes (in-patient or out-patient). A similar meta-analysis seems to be underway PMID: 34032747 DOI: 10.1097/MD.0000000000026099; this latter is limited to COPD patients, comparing out-patient rehabilitation vs home-based rehabilitation. The paper is very well written and enjoyable to read; the authors can be congratulated for this. The objectives are perfectly clear and the methods appropriate. This meta-analysis has been registered on PROSPERO and the description is consistent with the manuscript. I have only one comment for the authors: In the PROSPERO registration, the authors propose 2 hypotheses: Either home rehabilitation is superior to centre-based rehabilitation; or non-inferior?
---

	In my opinion, only one hypothesis can be retained. However, unless I am mistaken, this hypothesis is not reported in the manuscript. Depending on the hypothesis retained, the statistical analysis (meta-analysis) must be adapted (in particular it is necessary to set a non-inferiority limit) I suggest that the authors clarify this point and also specify the statistical models that will be used for the analyses.
--	---

VERSION 1 – AUTHOR RESPONSE

Reviewer 1: Dr. Adam Lewis, Imperial College London

1. Thank you to the authors for this submission. I read it with great interest. The need for home-based PR has become increasingly important in the context of COVID-19, and also the importance of structured availability of PR in lower-middle income countries is well stated. Your manuscript is brief and you have plenty more words to play with and so I have made some suggestions below which will hopefully help in you considering some amendments to your work.

Thank you.

2. The abstract is clear and details databases, PICO focus, quality assessment and research questions. It may be beneficial to add example search terms and types of studies included (? RCT only) within the abstract.

Thank you for your suggestion. We have updated the text in the Abstract on page 2, which now reads:

“The strategy is to search for ‘Chronic Respiratory Disease’ AND ‘Pulmonary Rehabilitation’ AND ‘Home-PR’, and identify relevant randomised controlled trials and controlled clinical trials”

Introduction

3. You need a paragraph in your introduction of similar research performed in the field already. In the introduction it may be worth referencing the recent Cochrane review in telerehabilitation [Cox NS, et.al. Cochrane Database Syst Rev. 2021;1:Cd013040] and maybe the work by Taito et al [Taito Set al.. Respir Care 2021;66(4):686-98], and Wuytack et al [Wuytack et al. Respirology 2018;23:272-83] illustrating the differences in your methods and highlighting/increasing the evidence for the gaps for you to address. There may be others too.

Thank you for suggesting we include these important reviews, two of which we had overlooked because they were published at or after the time we were finalising and submitting our protocol. We now have added some text in the introduction citing these papers and the gap that our review aims to fill. The text now reads:

“A Cochrane review of 65 studies (3822patients) has established the effectiveness of standard Centre-PR programmes in COPD,¹ with a sub-group analysis suggesting that PR delivered in a hospital centre may have a greater treatment effect than PR delivered in the community/home. Using the same definitions, three reviews (Wuytack et al,² Chen et al. 2020,³ and Neves et al⁴) included studies that compared PR delivered in different settings and both concluded that home/community-PR could be as effective as Centre-PR for people with COPD. Combining home and community services, however, overlooks the distinction between a community-based group supervised in person by a healthcare professional and a programme delivered to an individual in their own home. These reviews are also limited by disease (COPD only), although there is evidence that PR is of benefit in bronchiectasis⁵ and ILD.⁶ Taito in a scoping review also included people recovering from COVID-19.⁷

A recently published Cochrane review assessed the effectiveness and safety of telerehabilitation for people with CRDs when compared to Centre-PR or no rehabilitation.⁸ and concluded that primary or maintenance PR telerehabilitation achieved similar outcomes to Centre-PR. In this review, remote delivery of PR was defined by the use of telecommunications technology to deliver PR services to individuals or groups (either physical or virtual) in any location, including in the patient's home or at a healthcare centre. In contrast, in our review, the definition of HomePR is that the sessions are undertaken by an individual by themselves (though a family member may be involved) and typically at home. Apart from baseline and post-PR assessments,⁷ the patient does not attend a Centre (either a hospital Centre or a local 'satellite' Centre) and is not supervised face-to-face by a healthcare professional (though there may be remote communication from a healthcare professional for some or all of the session).

An additional distinction is that we defined Home-PR as comprising both exercise and at least one non-exercise component for a duration of not less than 4 weeks. This contrasts with other reviews^{2,7,8} that included exercise training programmes (i.e. without the non-exercise component that is normally included in Centre-PR.⁹) These reviews did not seek to identify components with greater impact on positive patient outcomes.

1. McCarthy B, et al. Pulmonary rehabilitation for chronic obstructive pulmonary disease. Cochrane Database Syst Rev 2015
2. Wuytack F, et al. Comparison of outpatient and home-based exercise training programmes for COPD: a systematic review and meta-analysis. *Respirology* 2018; 23:272-83
3. Chen YY, et al. Home versus centre-based pulmonary rehabilitation for patients with chronic obstructive pulmonary disease: a systematic review and metaanalysis. *TMR Integrative Medicine*. 2020;4::88888-e20012
4. Neves LF, Reis MHD, Gonçalves TR. Home or community-based pulmonary rehabilitation for individuals with chronic obstructive pulmonary disease: a systematic review and meta-analysis. *Cad Saude Publica* 2016; 32: e0008591
5. Bradley JM, et al. Physical training for bronchiectasis. *Cochrane Database Syst Rev* 2002
6. Dowman L, et al. Pulmonary rehabilitation for interstitial lung disease. *Cochrane Database Syst Rev* 2014.
7. Taito S, et al. Telerehabilitation in Subjects With Respiratory Disease: A Scoping Review. *Respiratory Care* 2021;66:686-98
8. Cox NS, et al. Telerehabilitation for chronic respiratory disease. *Cochrane Database of Systematic Reviews* 2021
9. Spruit MA, et al. An official ATS/ERS statement: key concepts and advances in pulmonary rehabilitation. *Am J Respir Crit Care Med* 2013; 188: e13-e64

4. A definition of what home-based PR is would be good within the main text of the review, considering it appears that some telehealth programmes will and will not be included in your table. Also, it sometimes can only be called home PR if a centre-based assessment is performed. This needs to be made clear.

Thank you for bringing this lack of clarity to our attention. Although, we provided the definition in table 1 on page 12, we now have put the definition in the introduction section (please see our response to #3). In addition, we have amended the definition of Home-PR to make it clearer. The definition of Home-PR in the table 1 on page 12 now reads:

“Home-PR”- the key criterion is that the sessions are undertaken by an individual by themselves (though a family member may be involved) and typically at home. Apart from baseline and post-PR assessments,¹ the patient does not attend a Centre (either a hospital Centre or a local ‘satellite’ Centre) and is not supervised face-to-face by a healthcare professional (though there may be remote communication from a healthcare professional for some or all of the session)”.

1. Taito S, et al. Telerehabilitation in Subjects with Respiratory Disease: A Scoping Review. *Respiratory Care* 2021; 66:686-98

5. List advantages and disadvantages of including all Chronic Respiratory Diseases as the population vs respiratory disease specific populations separately. From a clinical perspective it is likely all CRD patients included would be participating in a programme together so that is relevant and applicable.

However, methodologically it may cause some issues when comparing results. Will you perform a sub-group or sensitivity analysis per respiratory condition?

Thank you for your comment. We have added a few words in the introduction mentioning the importance of including all CRDs on page 5 (see our response to #3). We agree with you that some PR groups may include mixed conditions which will make it difficult to compare diseases. We have now amended the sentence about sub-group analysis on page 8 as follows:

“Subgroups may include high/low- and middle-income countries, CRD diagnosis (e.g., COPD, ILD, bronchiectasis, stable post-COVID lung disease, mixed CRD), severity as defined in internationally recognised guidelines,¹ intensity of intervention (number of weeks, sessions per week, workload, completion rate), and arrangements for supervision of the PR programme”

1. Global Initiative for Chronic Obstructive Lung Disease. 2021 Global Strategy for Prevention, Diagnosis and Management of COPD.

<https://goldcopd.org/2021-gold-reports>

Objectives

6. I really like the ambition of the multi-faceted objectives here, all of which are important. Good to define completion rate too. Does this mean that trials using a lower completion rate than this will be excluded from the analysis, like Holland et al [Holland AE et al. Thorax. 2017;72:57-65]. Alternatively you could lower your threshold for completion as has been the case in previous studies, and relates to the improvements seen after 4 weeks [Sewell et al. Thorax 2006;61:767-71].

Thank you. We agree about the importance of completion rate both a marker of fidelity, but also because improved completion is a possible mechanism by which Home-PR may work. We will not exclude trials with poor completion but if sufficient studies report completion rate we will undertake a sub-group analysis. We have now included this is our list of possible sub-group analyses on page 8 (see our response to #5 for the new text)

7. Ref 35: An updated 2020 PRISMA statement is available and recommended to follow on the PRISMA website and in this paper: The PRISMA 2020 statement: an updated guideline for reporting systematic reviews.

Thank you. The PRISMA reference (now ref 38) has been updated accordingly.

8. Line 50: In your search strategy please report that you will include disease specific search terms, even if you don't list them all, as you do in the Pubmed term table.

We have updated the sentence under Search strategy section on page 6, which now reads:

“We will develop a search strategy, including disease-specific search terms, and identify records through searching the following databases: PubMed, CINAHL, Cochrane, EMBASE, PeDRO, and PsycInfo (Appendix 1)”

9. Line 52: state hospital centres – community based approached are often in centres too (like leisure centres) and this may help differentiate.

We have now clarified on page 4 that ‘Typically PR is provided in hospital centres (Centre-PR)’

Search strategy:

10. Depending on your research questions it may be appropriate to include different study designs, so RCTs would be appropriate to study clinical effectiveness but other study designs wouldn't. Determining the completion rates and frequency or matrix of components may enable you to look at other research designs. For example Nolan et al(6) performed a propensity matched cohort study comparing completion rates which may be useful to consider but would currently be excluded. This may also be useful to consider:

<https://www.lungindia.com/article.asp?issn=09702113;year=2021;volume=38;issue=3;spage=211;epage=215;aulast=Priya>

You may also wish to follow up the authors of this paper to see if publication is planned:

https://www.atsjournals.org/doi/pdf/10.1164/ajrccm-conference.2021.203.1_MeetingAbstracts.A4148 Therefore, I think it is worth stating for each research question which study designs would be considered. I appreciate this information is in Table 1. But I think warrants reiterating specifically per research questions.

Thank you for these helpful suggestions and for sharing the useful links with us. Unfortunately, we cannot now look at alternative designs to answer questions 2 and 3 because, since we submitted the protocol to BMJ Open (in February 2021), we have progressed the work and completed the searches.

11. If you are interested in PR in stable-state COVID-19 patients you should include such terms in your search terms like COVID19 OR COVID-19 OR SARS-CoV-2. In the methods a description of what a chronic respiratory disease is may help justify COVID-19 as a chronic disease, because this could also be debated. See this article which shows

persistent changes at 12 months post COVID

[https://www.thelancet.com/journals/lanres/article/PIIS22132600\(21\)00174-0/fulltext](https://www.thelancet.com/journals/lanres/article/PIIS22132600(21)00174-0/fulltext)

However, individuals with COVID-19 may require a different PR due to different educational needs and exercise prescription potentially, also needing outcome measures to be validated, so I'm not sure results from such trials could be compared with other respiratory diseases. Perhaps a sub-group analysis?

Thank you for sharing this link with us. As explained in the previous response we have now completed the searches so cannot include additional terms. We agree that emerging evidence about long-COVID suggests that bespoke PR may be required (though whether that will be affordable in low-and middle-income countries is less clear). If we identify relevant papers, we will include COVID as a sub-group of CRD, but otherwise this will need to be a topic for a future systematic review.

12. Why are six reviewers required? Often only two required with a third to discuss discrepancies. Also, why in pairs and what does the training of the 100 randomly selected articles involve, How will this be random? This is interesting as I haven't read about this method of reviewing with such a large team and understand it will have its merits but may also have some limitations. These would be important to discuss so others could consider whether valuable for their reviews. If this method has been used elsewhere, please reference. If not, it may be novel, which isn't necessarily a bad thing.

We are a global multi-disciplinary team, funded by the RESPIRE Global Health Unit (<https://www.ed.ac.uk/usher/respire>), undertaking feasibility studies of PR in low-resource settings. The interest in Home-PR emerged from this feasibility work and representatives of the teams from Bangladesh, India and Malaysia opted (unfunded) to undertake this systematic review in their spare time. The main advantage of involving six reviewers is that it shares out the work (one-third of the total records will be assigned to each pair) and will allow us to complete the review in a timely manner and without over-burdening any individual. The six reviewers will work independently in pairs (as in the traditional model) so that all titles and abstracts are duplicate screened and disagreements resolved in discussion (involving the whole team as necessary). The main limitation is the potential for inconsistency, so before starting screening, 100 articles will be selected randomly from the total records by the study librarian and given to each pair to screen as a training exercise. Decisions will be discussed within the study team and operational rules clarified and agreed.

Whilst not normally needed in a fully funded review, this approach is not novel – indeed we have used this approach on a number of occasions where large numbers of 'hits' are expected.[e.g. Hanafi et al. *J Glob Health* 2021;11:04026; and Buelo et al. *Thorax* 2018;73:813–824] It is also a useful approach for unfunded reviews (e.g. student systematic reviews) where several second reviewers can be recruited to assist. We have now clarified this process in the methods revising the sentence on page 6 to read:

“Following training on 100 randomly selected records, six reviewers working in pairs (MNU and TJ, JPE and FT, DA and PJ) will duplicate screen titles and abstracts and identify potentially eligible studies”

Outcome measurement:

13. Provide some more justification for having more than 1 primary outcome measure. Most home-PR Trials previously have included a centre-based assessment and so will have a standardised evidenced based outcome measure (6MWT, ISWT, ESWT) [Holland et al. Thorax. 2017;72:57-65][Nolan et al. Thorax. 2019;74:996-8][Horton et al. Thorax 2018;73:29-36][Hansen et al. Thorax 2020;75:413-21][Tsai et al. Respiriology. 2017;22:699-707] so this may be a justification you can provide for choosing this as a primary outcome. Without such justification some clinicians may feel that such tests are often not used clinically in home-based PR programmes because they are difficult to perform and others may be preferred such as the Sit to Stand Tests. These however have not been widely validated for exercise prescription purposes.[Houchen-Wolloff et al. ERJ Open Res 2020;6] Although a centre-based assessment is deemed an essential component of a PR programme.[Holland et al. ATS Workshop Report. Ann Am Thorac Soc. 2021;18:e12-e29] You could add step tests and sit to stand tests (5repSTS, 1 minSTS) into your table of exercise capacity tests to allow for such home-based transferable outcomes to be analysed, should your definition of home-based or telehealth programmes including those without a centre-based assessment. Health related quality of life questionnaires can definitely be completed in the home or centre-based environments.

Thank you for these thoughtful comments. We have adopted similar outcomes to the existing reviews. For example, the recent Cochrane Review lists both functional exercise capacity and quality of life as primary outcomes (as well as dyspnoea and adverse events). These reflect our definition of PR as an intervention that comprises both exercise and at least one non-exercise component.

We state (in Table 1) that we would be looking for validated instruments and hence specify SGRQ and CRQ as preferred quality of life instruments, and 6MWT, ISWT, ESWT as our preferred functional exercise outcomes. Although most Home-PR programmes undertake Centre-based assessment,¹ we agree that some Home-PR studies may use 'step tests' and 'sit to stand' tests so have now added these as possible outcomes in Table 1:

"We will also include step tests and sit-to-stand tests that are sometimes used in Home-PR assessments²"

We have also clarified that many Home-PR interventions undertake Centre-based assessments in our definitions (Table 1) and also in the introduction on page 5:

"Apart from baseline and post-PR assessments,¹ the patient does not attend a Centre"

1. Taito S, et al. Telerehabilitation in Subjects with Respiratory Disease: A Scoping Review. *Respiratory Care* 2021; 66:686-98
2. Holland AE, et al. Home-based or remote exercise testing in chronic respiratory disease, during the COVID-19 pandemic and beyond: a rapid review. *Chronic Respiratory Disease* 2020;17:1479973120952418

All three of your outcomes are really important and clinically relevant Outcome one:

14. What is the statistical test or final outcome expected? pooled mean differences, standardised mean differences. Also state what models you are considering for meta-analysis.

Thank you for highlighting the need for additional information about the analysis. We have added a paragraph with further details under the Data analysis section on page 8. The text now reads:

"For homogenous data from RCTs, we will perform a pooled quantitative synthesis using an inverse variance method and a random-effects model in the meta-analysis. We will consider pooled mean differences if the same outcome measurement tool is used in the included RCTs. However, if outcome measurement tool varies among trials, we will consider standardised mean differences. We will use Review Manager software (RevMan 2020, version 5.4.1) to perform meta-analysis".

15. Consider how you will account for missing data in trials from reporting quality (possibly in your secondary outcome measures this may occur).

Under the 'Data management and extraction' section, we had already mentioned that we will contact authors for any missing data. We now have added a sentence on page 7. The text now reads:

"We will contact authors for any missing data. If this is not possible and the missing data seem to introduce serious bias, we will perform sensitivity analysis of the impact of including such studies in the overall assessment of results".

16. What software will you use to perform the meta-analysis?

RevMan 2020, version 5.4.1. Please see response #14 for how we now report this under Data analysis

17. Will usually care, and centre-based control group study outcomes be pooled?

No. We will undertake separate analyses comparing Home-PR firstly with Usual Care and then with Centre-PR. We have clarified this on page 8 which now reads:

"We plan to undertake meta-analysis for the primary outcomes and some secondary outcomes (e.g., HRQoL, dyspnoea, muscle fatigue, exacerbations, hospitalisations), comparing Home-PR firstly with Usual care and then with Centre-PR"

Outcome 2:

18. This paper may be a good read regarding how to explore the value of various components within a PR intervention [Machado et al. Eur Respir Rev 2020;29:]

Thank you for highlighting this interesting paper, however, it is beyond the scope of this review to perform a network meta-analysis.

19. Outcome 3: How will this be analysed? Will an overall mean completion rate be presented or will you compare this means with a statistical test? chi-squared test?

Our scoping work suggests that few trials report completion rate, and those that do often confuse it with trial attrition. We therefore do not anticipate having sufficient data to undertake statistical testing. We therefore plan a narrative approach to analysis. We have added completion rate as a potential sub-group meta-analysis, in the (unlikely) event that sufficient trials report this with sufficient clarity. In addition to clarifying our analysis plan for objective 3, we have taken the opportunity to align with the recent Cochrane review that has adopted a 70% threshold.¹ The amended text on page 8 now reads:

"We will use a narrative approach to synthesise completion in Home-PR and Centre-PR groups. If sufficient studies report completion rates we will consider a sub-group analysis based on the threshold of 70% completion".

1. Cox NS, et al. Telerehabilitation for chronic respiratory disease. Cochrane Database of Systematic Reviews 2021

Risk of Bias:

20. Will all studies be included in quantitative analyses regardless of quality?

How will you assess the risk of bias in non-randomised controlled trials? Consider ROBINS-I [Sterne et al. BMJ. 2016;355:i4919]

Thank you for raising these issues. We will include all studies in our primary analysis, but take into account the risk of bias of the studies when considering the intervention effects. If there are sufficient studies we may undertake sensitivity analyses omitting studies at high risk of bias

In addition, we have now have added the possibility of non-randomised controlled trials in the 'Risk of bias assessment' section on page 7. The text reads:

"To assess risk of bias of CCTs, we will use the 'Risk of Bias in Non-randomised Studies of Interventions' (ROBINS-I) tool.⁴¹ We will include all studies in our primary analysis but take into account the risk of bias of the studies when considering the intervention effects. If there are sufficient studies we may undertake sensitivity analyses omitting studies at high risk of bias".

1. Sterne JA, et al. ROBINS-I: a tool for assessing risk of bias in non-randomised studies of interventions. *BMJ* 2016;355:i4919

PPI

21. "Patients are involved in the with the RESPIRE programme of work on pulmonary rehabilitation." Needs rewording

Thank you for spotting this typo. We have corrected this and the text (on page 9) now reads:

"Patients who are involved in the RESPIRE programme of work on PR-have endorsed the importance of Home-PR for improving accessibility to rehabilitation. They will be involved in interpreting the findings and the implications for intervention development and the overall programme of work".

Other

22. The statement in your definition for home-PR requires a little more justification: "The patient does not attend a Centre, is not directly supervised by a healthcare professional (though they may be monitored remotely for some or all of the session)." In my experience telehealth PR is often lead by a clinician remotely, which implies supervision rather than monitoring. Is there a difference with monitoring that means no direct communication is made between the clinician and patients perhaps? In the definition of Centre-based PR you state that telehealth programmes are included in this arm too, but if patients attend a centre. Why would they need to attend a centre in order to do telehealth? Is the main important difference here that patients are exercising in a group? This addition may help add clarity too

Thank you for drawing our attention to the lack of clarity in our definition of Home-PR

- We agree that remote telerehabilitation often uses satellite centres which are supervised by a healthcare professional. We are arguing that this is 'Centre-PR' because a) it is delivered to a group and b) because there is supervision with the physical presence of a healthcare professional (albeit often less qualified than the senior staff at the main hospital centre)
- Yes, we had used the word 'monitoring' to imply something less intensive than a healthcare professional 'supervising' a group face-to-face, but recognise that we had not made that clear. We have now avoided the word 'monitoring' in our definition.
- 'Telerehabilitation' (as in the Cox Cochrane review¹ implies a mode of communication and does not define the group or individual nature of the intervention.

The key distinction that we are making is that Centre-PR patients are exercising in a group (which might be the hospital Centre or a local 'satellite' centre typically with a healthcare professional in attendance), and Home-PR patients are exercising as individuals without the benefit of peer encouragement or the physical presence of a healthcare professional. This argument is now built more clearly in response to your earlier comment (see #3 above). Our definition now reads:

"In our review, the definition of Home-PR is that the sessions are undertaken by an individual by themselves (though a family member may be involved) and typically at home. Apart from baseline and post-PR assessments,[ref Taito] the patient does not attend a Centre (either a hospital Centre or a local 'satellite' Centre) and is not supervised faceto-face by a healthcare professional (though there may be remote communication from a healthcare professional for some or all of the sessions).

1. Cox NS, et al. Telerehabilitation for chronic respiratory disease. Cochrane Database of Systematic Reviews 2021

Hope this helps and good luck in your revision.

Many thanks for you considered and detailed comments

Reviewer 2: Dr. Aimee Layton, Columbia University Medical Center

This manuscript describes the protocol and rational for performing a thorough review of the current literature on the effectiveness of home pulmonary rehabilitation compared to center based. The rational, methods and discussion are clear and well supported. I have no major comments.

Thank you very much.

Reviewer 3: Dr. Jean BOREL, INSERM U1042

Thank you for inviting me to review this manuscript. Dr Uzzaman is currently conducting a systematic review with meta-analysis to evaluate the effectiveness, completion rates and components of home-based versus centre-based rehabilitation programmes (in-patient or out-patient). A similar meta-analysis seems to be underway PMID: 34032747 DOI: 10.1097/MD.0000000000026099; this latter is limited to COPD patients, comparing out-patient rehabilitation vs home-based rehabilitation.

There is a limited description of the intervention being investigated in this review, but our understanding is that the Shi review is including 'unsupervised' home-PR, and excluding these with remote contact (such as telephone calls) which would be included in our review.

The paper is very well written and enjoyable to read; the authors can be congratulated for this. The objectives are perfectly clear and the methods appropriate. This meta-analysis has been registered on PROSPERO and the description is consistent with the manuscript.

Thank you.

I have only one comment for the authors: In the PROSPERO registration, the authors propose 2 hypotheses: Either home rehabilitation is superior to centre-based rehabilitation; or non-inferior? In my opinion, only one hypothesis can be retained. However, unless I am mistaken, this hypothesis is not reported in the manuscript. Depending on the hypothesis retained, the statistical analysis (meta-analysis) must be adapted (in particular it is necessary to set a non-inferiority limit) I suggest that the authors clarify this point and also specify the statistical models that will be used for the analyses.

Thank you for your comments.

Firstly, our two hypotheses are separate and refer to different comparators:

1. Home-PR is superior to Usual Care. This will be a meta-analysis using Standard Mean Differences (SMDs) as we anticipate a range of outcomes (e.g. 6-MWD, ISWT, ESWT)
2. Home-PR is not inferior to Centre-PR. This will also be investigated using a meta-analysis using Standard Mean Differences (SMDs) to include the range of outcomes (6-MWD, ISWT, ESWT)

In response to Reviewer 1, comment 17 we have now clarified that these are separate analyses with different comparators. The text on page 8 now reads:

"We plan to undertake meta-analysis for the primary outcomes and some secondary outcomes (e.g., HRQoL, dyspnoea, muscle fatigue, exacerbations, hospitalisations), comparing Home-PR firstly with Usual care and then with Centre-PR"

Secondly, apparent discrepancy with the PROSPERO protocol

We recognise that use of the terminology 'non-inferior' in our hypothesis in the PROSPERO submission raises specific expectations about defining non-inferiority margins typically based on Minimum Clinically Important Differences (MCIDs). This, however, is only possible in meta-analyses of studies using the same outcome measures. Since submitting the PROSPERO protocol, we have completed the searches and it is clear that (for example) functional exercise capacity is reported using multiple measures and we will be converting outcome measures into standardised mean difference (SMD) for analysis. We have discussed your question with a statistician who confirms that 'there isn't a way to infer a clinically meaningful non-inferiority margins on the SMD'. We will therefore use a standard meta-analysis for our comparison between Home-PR and Centre-PR (which we note is the approach taken by the recent Cox et al Cochrane Review,¹ presumably for the same reason). If there are enough studies using the same measure for one of the outcomes, we will define the non-inferiority margin as the minimum clinically important difference.

We have now explained this in the data analysis section on page 8 which now reads:

"Our hypothesis is that Home-PR is non-inferior to Centre-PR, but a clinically meaningful non-inferiority margin cannot be inferred using SMDs. If sufficient studies use the same measure for functional exercise capacity or healthrelated quality of life, we will define the non-inferiority margin as the minimum clinically important difference".

We would like to thank your reviewers for their considered comments, which have helped us improve our paper. Attached is an updated version of the manuscript with highlighted changes which has been seen and approved by all the co-authors. We believe we have responded to all the points made and hope the manuscript now meets with your approval. Please feel free to contact us if you require any further clarification.

VERSION 2 – REVIEW

REVIEWER	Lewis, Adam Imperial College London, Department of Respiratory Epidemiology, National Heart and Lung Institute
REVIEW RETURNED	08-Sep-2021

GENERAL COMMENTS	Dear Authors, Thank you very much for all of your responses to the reviewer comments. I found they were thorough. Good discussions were provided regarding adding suggested revisions from the reviewers, and also reasons why you were unable to accommodate others into your article which again I found were well justified. This remains an exciting, clinically relevant systematic literature review and warrants publication in my opinion I look forward to reading your systematic review once completed. Best of luck with it!
--

REVIEWER	BOREL, Jean INSERM U1042, HP2 Laboratory
REVIEW RETURNED	06-Sep-2021

GENERAL COMMENTS	The authors answered my questions, thank you
--